# An Experimental Ultrasound System for Qualitative Tomographic Imaging

**DOI:** 10.3390/s22207802

**Published:** 2022-10-14

**Authors:** Michele Ambrosanio, Stefano Franceschini, Maria Maddalena Autorino, Fabio Baselice, Vito Pascazio

**Affiliations:** 1Dipartimento di Scienze Motorie e del Benessere, Università Parthenope, Centro Direzionale di Napoli, 80143 Napoli, Italy; 2Dipartimento di Ingegneria, Università Parthenope, Centro Direzionale di Napoli, 80143 Napoli, Italy

**Keywords:** ultrasound tomography, ultrasound systems, tomographic imaging, MIMO systems, inverse scattering, coherent imaging, biomedical imaging

## Abstract

The advancement of new promising techniques in the field of biomedical imaging has always been paramount for the research community. Recently, ultrasound tomography has proved to be a good candidate for non-invasive and safe diagnostics. In particular, breast cancer imaging may benefit from this approach, as frequent screening and early diagnosis require decreased system size and costs compared to conventional imaging techniques. Furthermore, a major advantage of these approaches consists in the operator-independent feature, which is very desirable compared to conventional hand-held ultrasound imaging. In this framework, the authors present some imaging results on an experimental campaign acquired via an in-house ultrasound tomographic system designed and built at the University of Naples Parthenope. Imaging performance was evaluated via different tests, showing good potentiality in structural information retrieval. This study represents a first proof of concept in order to validate the system and to consider further realistic cases in near future applications.

## 1. Introduction

Non-invasive imaging is one of the major topics involved in biomedical engineering research. The goal is to produce quantitative and/or qualitative maps of inner human tissues in order to help and guide doctors in diagnoses or surgeries [1].

During recent years, most biomedical images were provided by technologies based on ionizing radiations, such as X-ray computed tomography (CT), γ-ray single photon emission computed tomography (SPECT) and positron emission tomography (PET). The use of these forms of radiation limits the time exposure of patients as the radiation is unsafe for their health. For this reason, magnetic resonance technology progressively substituted CT, SPECT and PET exams where possible [1]. Magnetic resonance imaging (MRI) provides high spatial resolutions in biomedical images at the expense of long-time acquisitions and the high cost and size of the apparatus.

In this framework, an easy and cheap solution is represented by ultrasound (US) imaging [2]. In its conventional approach, US imaging suffers for the operator-dependant nature of the exam and images are corrupted by the speckle noise that degrades the quality of the results [3,4,5].

In order to combine the benefits of ultrasound waves and the stability of tomographic approaches, ultrasound tomography (UST) can represent a good candidate for non-invasive imaging. In UST, pressure fields are generated and measured by US sensors located outside a region of interest (ROI), and maps of compressibility, density and attenuation of the objects located in the ROI can be provided [6,7].

The configuration of sensors surrounding the object under investigation is not relatively new; indeed, many research groups proposed their configurations, especially for breast imaging applications since the 1980s [8,9]. In early prototypes, bi-static configurations were considered with a single pair of transmitter and receiver being adopted. Recently, UST systems present multiple-input and multiple-output (MIMO) configurations where several US transducers are considered [10,11,12,13,14,15]. Moreover, this approach might be also adopted in hybrid systems where UST is considered in combination with other approaches, among which microwave tomography seems very promising [16,17,18].

Several scientists have been working on ultrasound tomography by focusing on different aspects that span from the principles of ultrasonic reflectivity to the analytical expressions of the point-spread function and to the data acquisition protocol in order to speed up processing times [19,20,21]. However, for high-quality images, fast acquisition techniques are paramount to avoid movement artifacts that have a strong impact on imaging performances.

Some of the main challenges related to the clinical use of UST can be summarised as follows:The need for a large number of US sensors in order to obtain a proper resolution, which must be low cost and have very similar characteristics (in terms of operating frequency band and radiation pattern). Furthermore, the size of the sensors is paramount, since smaller sensors are desirable for the radiation pattern, but they result in low pressure levels, which results in a low signal-to-noise ratio.The acquisition time must be very short to avoid movement artifacts and to allow sub-millimeter resolutions.Due to the attenuation, the received signal at the transducer’s location can be very low with respect to the transmitted one, which generates large dynamic ranges (this aspect is quite limiting, especially for diagnostic purposes in the biomedical field).Proper processing based on coherent imaging is paramount to obtain high signal-to-noise and signal-to-artifact ratios at good resolutions. Nevertheless, this has a strong impact in the accuracy of evaluating the transducer’s position, which must be known with an uncertainty lower than millimeter fractions at a few MHz frequency. Thus, the precise calibration of sensors position is compulsory.

A paramount aspect of UST is also related to the reconstruction algorithm. The presence of multiple interactions and the unavoidable presence of noise render the considered inverse scattering problem’s (ISP) solution difficult, since it is non-linear and ill-posed [22,23]. Therefore, the solution is unstable, leading to the need of proper a priori information or regularization strategies for proper inversions [24].

The first experimental demonstrations of ultrasonic tomography were performed in the early 1970s [25]. Due to the limited computational resources of that time, the first imaging attempts were based on simplifications of the wave equation ruling the propagation and scattering phenomena (e.g., constant density, lossless background medium, etc.). A very significant simplification is based on *straight-ray propagation* [26,27], which is a suitable approximation in the biomedical field when imaging soft tissues [28,29]. A widely adopted approach in this field is the filtered backprojection algorithm [30,31]. Further approaches include the simultaneous algebraic reconstruction technique (SART) [32] and reconstruction methods for fan-beam tomography [33].

Nevertheless, more refined approximations can be adopted rather than the simple straight-ray model. As a matter of fact, refraction phenomena can occur when US waves propagate through soft tissues, which reduces the resolution capability of simple imaging approaches. Thus, it is important to incorporate these reflection effects in the ray-based US tomography [34,35]. Among these *ray-linking* approaches, which can be computationally expensive, a particularly interesting class of algorithms consists of using graph theory methods to find the best path-connecting trasmitter–receiver pairs [36,37]. The main drawback of the refraction-corrected UST is related to the fact that the spatial distribution of the refraction index has to be known, which is the target of the considered estimation procedure. Therefore, the imaging problem at hand becomes nonlinear with respect to the speed of sound distribution, and more sophisticated approaches are required.

Among them, the framework of *diffraction tomography* has been particularly interesting. This approach was formally introduced in the US community in the late 1970s by Muller et al. [38,39], even though its theoretical formulation dates back prior to ray-based acoustic tomography [40]. In this framework, it is possible to obtain a closed-form solution via adopting a first-order Born approximation, which is based on the hypothesis that the field scattered by targets located in the imaging domain is negligible; thus, the nonlinear problem at hand becomes linear [41,42,43]. Nevertheless, other linearization strategies can be adopted further to Born approximation. For instance, the first-order Rytov approximation was also applied to obtain analytical solutions to the wave inversion problem [42,44].

Relatively to diffraction tomography, conventional approaches exploits the spatial diversity of transmitter–receiver pairs as degrees of freedom to perform imaging, but frequency diversity can also be exploited as a further degree of freedom to collect more independent information [45,46]. Moreover, interpolation methods in the frequency domain are also desirable for efficient UST since they obtain good results in terms of reconstruction quality after increasing sampling densities by using zero-padding [47].

Even though previous approaches are quite quick and convenient for retrieving geometrical and structural information relative to the scenario under test [48], more advanced and complete methods are required for the retrieval of the quantitative properties of considered tissues [49]. Thus, these approaches, known as *full wave* inversion methods, allow quantitative recoveries but are characterized by a higher computational cost than diffraction tomography. Most of these latter inverse scattering techniques are iterative and include the method of alternating variables (also known as iterative Born method, [50]), the Newton-type methods (or distorted Born iterative method, [51,52,53]) and conjugate gradient methods [54,55,56,57,58], among which the contrast source inversion is a notable example [49]. Nevertheless, other iterative algebraic methods can be adopted, such as Kaczmarz’s method, which successively refines the current estimate by performing orthogonal projections on hyperplanes related to a matrix operator [59], or eigenfunction methods, which rely on the eigenfunctions of the far-field operator relating an incident plane wave from a certain angle to the far-field scattering behaviour observed at another angle [60].

In this paper, an in-house ultrasound tomographic prototype is presented. The system consists of 22 US transducers (one transmitter and 21 receivers), and the angular variability is obtained by a mechanical rotation of the ROI guaranteed by a high-precision rotating table. For these tests, an air-matched system was considered. In these scenarios, basically, only structural information can be retrieved due to the high contrast between the acoustic properties of the targets under investigation and the background medium. It is worth noticing that the proposed system, conversely from the one proposed in [6,61], consists of a multistatic configuration composed of physical US transducers that surround the imaging domain, allowing the collection of a higher amount of (possibly) independent information with respect to the system proposed in [12]. As a matter of fact, this latter configuration is characterised by more transducers that partially surround the imaging domain, but despite their angular movement, does not allow the collection of as much spatial information as the proposed system can. Moreover, the considered prototype exploits a 40 kHz monochromatic sinusoidal signal. Thus, the cost of US transducers, cables and the overall acquisition can be very limited.

More in detail, due to the high contrast between the object under investigation and the background medium, it is possible to consider a linear approximation to retrieve the support information related to the target’s location and extension, thus providing a qualitative map of the considered scenario.

The reminder of the paper discusses the following: Section 2 discusses the mathematical formulation of the considered problem. Next, in Section 3, a brief presentation of the proposed UST system is provided. Section 4 contains some details regarding the experiments and related qualitative recoveries, while some conclusions are provided at the end of the paper in Section 5.

## 2. Mathematical Formulation

For the sake of simplicity, a two-dimensional (2D) scenario will be considered in the following section, as shown in Figure 1. Under this hypothesis, the integral equation governing the scattered field is a function of the compressibility, attenuation and density of the objects under investigation and of the background medium in which the signal is propagating.

More in detail, the attenuation parameter determines the decay of the amplitude of the propagating US signal and has an important role in medical imaging since it can vary significantly according to the types of tissue [62,63,64].

By considering an homogeneous background, the resulting scattered field equation can be written as follows (omitting the frequency dependence and the transmitter location) [65]:(1)$$\begin{array}{cc} p_{scat}(r) & =k_{b}^{2}{\;\int \int \;}_{\Omega }g(r,r^{\prime })[\chi_{1}(r^{\prime })-j\frac{2\delta \hat{\alpha }\left(r^{\prime }\right)}{k_{b}}]p_{tot}\left(r^{\prime }\right)dr^{\prime }\\ \; & +{\;\int \int \;}_{\Omega }g\left(r,r^{\prime }\right)\nabla \cdot [\chi_{2}\left(r^{\prime }\right)\nabla p_{tot}\left(r^{\prime }\right)]dr^{\prime }\\ \; & -{\;\int \int \;}_{\Omega }g\left(r,r^{\prime }\right)(\left[\chi_{2}\left(r^{\prime }\right)+1\right]\alpha^{2}\left(r^{\prime }\right)-\alpha_{b}^{2})p_{tot}\left(r^{\prime }\right)dr^{\prime }\;,\\ \; & \;r\in \Gamma \;, \end{array}$$
where $p_{scat}$ and $p_{tot}$ are, respectively, the scattered and the total pressure fields. In this formulation, the scattered pressure field is defined as the difference between the total and the incident pressure fields, e.g., $p_{scat}=p_{tot}-p_{inc}$; $k_{b}=\frac{\omega }{c_{b}}$ is the lossless background wavenumber; ω is the angular frequency; $c_{b}$ is the propagation speed in the background. Mathematical notation g(·,·) refers to the background Green’s function related to imaging domain Ω, which is defined as follows:(2)$$g\left(r,r^{\prime }\right)=-\frac{j}{4}H_{0}^{\left(2\right)}\left(k_{b},\left|r-r^{\prime }\right|\right)\;,$$
in which $H_{0}^{\left(2\right)}$ is the zeroth-order Hankel function of the second type, and *j* is the imaginary unit.

The two quantities $\chi_{1}\left(r\right)$ and $\chi_{2}\left(r\right)$ represent the contrast function for the compressibility and density defined as:(3)$$\chi_{1}\left(r\right)=\frac{\kappa \left(r\right)-\kappa_{b}}{\kappa_{b}}\;,\;\chi_{2}\left(r\right)=\frac{\rho^{-1}\left(r\right)-\rho_{b}^{-1}}{\rho_{b}^{-1}}\;,$$
where κ and ρ are, respectively, the compressibility and density and the subscript *b* refers to the same quantities related to the background. The quantity $\delta \hat{\alpha }\left(r\right)$ in Equation (Equation 1) is a function involving both contrasts defined in (Equation 3) and is equal to the following:(4)$$\delta \hat{\alpha }\left(r\right)=\alpha \left(r\right)\sqrt{\left[\chi_{1}\left(r\right)+1][\chi_{2}\left(r\right)+1\right]}-\alpha_{b},$$
in which α and $\alpha_{b}$ are inhomogenous and background attenuations, respectively.

For the scope of this communication, the data equation can be particularized in the case of scatterers possessing $\alpha \left(r\right)>>\alpha_{b}$, which leads to the following equation [66]:(5)$$p_{scat}\left(r\right)\approx {\;\int \int \;}_{\Omega }g\left(r,r^{\prime }\right)I\left(r^{\prime }\right)p_{tot}\left(r^{\prime }\right)dr^{\prime }=L[I\left(r\right)]\;,\;r\in \Gamma \;,$$
where the following is the case.
(6)$$I\left(r\right)=\alpha^{2}\left(r\right)[\chi_{2}\left(r\right)+1]-\alpha_{b}^{2}.$$

The aim of the proposed analysis is to find a qualitative and stable estimate (support estimation) of the targets located in the imaging domain, i.e., a map of the qualitative indicator $I\left(r\right)$. To this scope, a linear approximation is exploited by assuming $p_{tot}\left(r\right)\approx p_{inc}\left(r\right)$, also known as Born approximation (BA), and an estimate of the targets location is obtained via a truncated singular value decomposition (TSVD). Thus, exploiting the TSVD of operator L, which defines triplet ${\left\{\sigma_{n},u_{n},v_{n}\right\}}_{n=0}^{N}$, a formal solution to Equation (Equation 5) can be formulated as follows: (7)$$\begin{array}{cc} I\left(r\right)=\sum_{n=0}^{N} & \frac{1}{\sigma_{n}}\langle {\tilde{p}}_{scat},v_{n}\rangle u_{n}\left(r\right)\;, \end{array}$$(8)$$\begin{array}{cc} I_{n}\left(r\right) & =\frac{I(r)}{\max\left(I\right)}\;, \end{array}$$
in which 〈·,·〉 represents the scalar product, ${\tilde{p}}_{scat}$ represents the noisy data, *N* is the truncation index and max (·) is the maximum evaluation operator for the normalisation of the indicator in Equation (Equation 7).

It is worth noting that imaging operator L is compact, meaning that the corresponding inverse operator is not continuous. This aspect characterises most inverse problems that are known as “ill-posed” and more in detail as *ill-conditioned*. This aspect strongly impacts on reconstruction performance since even a limited amount of noise on the data can lead to false solutions. To mitigate this issue, the sequence of singular values $\sigma_{n}$ in Equation (Equation 7) is truncated [24].

## 3. System Overview

The UST system, for which its block diagram is reported in Figure 2, is mainly composed of a wooden circle where 22 US transducers are placed. Such components are responsible for converting voltages signals into acoustic pressure waves and vice versa; moreover, the system is also composed of a waveform generator exciting the transmitter, an analog-to-digital converter for the digitisation of the signals, a rotating table hosting the objects to be imaged and, finally, a computer for managing the acquisition, signal processing and image-formation steps.

More in detail, the region of interest is surrounded by 1 US transmitter (Tx) and 21 receivers (Rx) SensComp 40LT16 and 40LR16, respectively. For both Tx and Rxs, the operating frequency range is the band [38, 42] kHz (−6 dB band) with a radiating pattern of approximately ±30 at −6 dB in both vertical and horizontal planes. All sensing elements are placed on a wooden ring with radius of 17.5 cm. The angular distance between every transducer is approximately 16.36. A picture of the measurement ring is shown in Figure 3. The transmitter is connected to a waveform generator (Agilent Technologies 33220A), producing a monochromatic 40 kHz sine wave. By virtue of this operating choice, the in-air wavelength is approximately 8.5 mm. All transducers signals are acquired by an analog-to-digital converter (National Instruments, PCI-6251). The sampling frequency was fixed at 100 kHz while the precision is 16 bits.

In order to obtain more information, the view angle of the region of interest (ROI) is changed by means of a rotating table on which the targets are located, performing measures of the pressure field for 308 different angles. The acquisition process consists of fixing a ROI position and transmitting the excitation signal, acquiring a 0.5 s signal related to both Tx and Rxs transducers, rotating the table and repeating the acquisitions until all angular positions are covered. This entire process lasts approximately 4 h.

Once acquired, the signals must be properly processed. In particular, the so-called “beating process” is used, which consists of three steps. In the first one, both received and transmitted signals are reported in their analytic form by using the Hilbert transform [67]. The second step provides a multiplication between these two analytic signals (the transmitted one is conjugated). In the last step, a time average operation is computed by obtaining a complex number. Finally, a 308 × 21 complex matrix (angular views × receivers number) is obtained.

The acquisition protocol is divided in two steps: In the first one, a measure of the pressure waves related to the camera-void situation is performed. In this manner, the “*pvoid*” matrix is obtained. Similarly, the matrix “*ptot*”, e.g., the matrix of pressure waves when the object is placed inside the region of interest, is evaluated. Finally, the difference *ptot−pvoid* provides scattering matrix “*pscat*”.

The entire acquisition process (table rotation and data acquisition) is managed by a LabVIEW VI code, while the signal processing and image formation are carried out by MATLAB scripts.

## 4. Numerical and Experimental Results

In order to test the reliability of the proposed system in non-invasive imaging applications, tests with different objects are proposed. All considered objects are z-oriented cylinders with extremities that are far from the measurement plane in order to neglect side scattering effects, and they are driven into a simplified 2D geometry with an axial symmetry.

In the following, both numerical simulations and experiments will be considered for carrying out a performance assessment analysis. For the sake of clarity, two measurement configurations are taken into account, as reported in Figure 4. In the first case (Figure 4a), 308 US receivers and a single US transmitter was adopted, and for each scenario, data collection is repeated for 308 different equally spaced view angles (which corresponds to an angular step of $\theta_{\mathrm{rot}}=1.165{}^{\circ}$). Conversely, from the former scheme, in the measurement configuration of Figure 4b, only 21 receivers are adopted but they still cover 308 equally spaced view angles. Thus, with the aforementioned schemes, a total of 308×308 = 94,864 data are collected for the former configuration, while a total of 21×308=6468 data are collected in the latter case, i.e., approximately 7% with respect to the full-data case.

Regarding the numerical simulations, simplified two-dimensional (2D) scenarios corresponding to the ones adopted in the experimental campaign were considered and carried out via the k-Wave Matlab toolbox [68]. This toolbox exploits k-space pseudo-spectral time domain simulations for solving coupled first-order acoustic equations for both homogeneous and heterogeneous media. The parameters adopted in all simulations are reported in Table 1.

With respect to the experimental campaign, three different test cases were proposed. Figure 5a–c report some sketches for the considered scenarios. It is worth noting that for all real experiments, due to the proposed imaging system, data collection was performed by adopting the measurement configuration illustrated in Figure 4b.

The first scenario (Figure 5a) involves a single metallic cylinder with a 4 mm diameter placed at a 2 cm distance far from the ROI’s centre (with respect to the centre of the target). For a 10 × 10 cm^2^ imaging region, the qualitative reconstructions of this first example by exploiting Equation (Equation 8) are shown in Figure 5d,g,j, with the reference target position and extension defined by the red dash-dot line. More specifically, Figure 5d,g show the results of the corresponding numerical inversions obtained by processing the simulated numerical data collected with different measurement configurations, i.e., the ones shown in Figure 4. Finally, Figure 5j illustrates the processing of the experimental data, for which its collection was carried out via adopting the measurement configuration reported in Figure 4b.

It is easy to observe that both the object’s location and dimension are perfectly retrieved in Figure 5d, and this is also due to the higher amount of collected data (i.e., 308 receivers and 308 view angles; the measurement configuration is shown in Figure 4a). Conversely, the recovery performance slightly worsens in the case reported in Figure 5g since fewer data (i.e., 21 receivers and 308 view angles) were used due to the adopted measurement configuration (Figure 4b). It is worth noting that this data reduction process generates a “ring” effect visible in the retrieved image. Lastly, Figure 5j illustrates the support estimation obtained via Equation (Equation 8) in the case of experimental data processing (measurement configuration of Figure 4b). It is easy to observe that despite the unavoidable presence of noise, it is still possible to retrieve the correct target location and dimension.

The second case deals with two metallic 4 mm diameter cylinders arranged as shown in Figure 5b, with the two cylinders located at a 1 cm distance from each other. As for the single metallic cylinder case, some numerical simulations in the two different measurement configurations in Figure 4 were considered, and corresponding target support estimations are shown in Figure 5e,h. Moreover, in this case, the results are accurate, confirming the single-cylinder observations. Finally, Figure 5k shows the recovery obtained by processing the experimental data. Again, the approach allows one to correctly locate the targets with a sightly underestimation of their extension. Nevertheless, it is possible to note that the two objects are correctly localised and identified as separate, proving good spatial resolutions.

The last case considered herein is related to a hollow cardboard cylinder with a 3 cm diameter and 1.5 mm thickness. In this case, the centre of the object was placed at approximately 2.5 cm from the centre of the ROI, as shown in Figure 5c. This testbed is particularly interesting since it deals with a larger, hollow object that is particularly challenging by adopting a linear approximation for the target support estimation due to the highly nonlinear interactions related to multiple scattering phenomena. As a matter of fact, the adopted measurement configuration has a stronger impact in this case rather than for the small metallic cylinders, as proved by the results reported in Figure 5f,i, and the former shows improved results due to the higher amount of data used to perform processing. Finally, Figure 5l reports the recovery obtained by processing experimental data. As mentioned before and, conversely, from the small-metallic-cylinders case, here, the target is substantially larger with respect to the wavelength, and a linear model is assumed (Equation (Equation 5)), which justifies the presence of the ring effect in the image, as also confirmed by the recovery in Figure 5i, which also is affected by the same artifact.

With the aim of carrying out a performance assessment analysis in the case of limited data, some experimental support estimations of the previously presented test cases of Figure 5a–c are reported in Figure 6. More in detail, full data processing is shown in Figure 6a,d,g, i.e., by adopting the data collected at all 21 receivers for all 308 viewing angles. Then, support estimations were repeated by considering fewer view angles, i.e., 154 for Figure 6b,e,h and 77 for Figure 6c,f,i. It is worth noting that despite the considerable data reduction (up to 75% for the 77 view angle case), the recoveries related to smaller objects, i.e., the metallic cylinders, still preserve good quality. Conversely, a higher impact of the data reduction can be observed for the hollow–cardboard–cylinder case, for which a 75% reduction in data strongly limits the correct estimation of target location and extension; in this case, more appropriate inversion models and, eventually, regularization strategies are required in order to obtain improved recovery performances.

## 5. Conclusions

In this paper, an in-house ultrasound tomography prototype was proposed and investigated. The system, exploiting a circular array of acoustic transducers, is capable of providing information about objects placed in a region of interest in a safe and non-invasive investigation modality.

The system’s imaging performances were evaluated using different test cases. This analysis confirmed the potentiality of the system; indeed, the position and size of the objects were correctly retrieved. Moreover, the system is characterised by good spatial resolution that is able to distinguish targets that are close. In addition, the hollow nature of the objects was correctly retrieved as well.

It is worth recalling that the reconstruction problem is strongly non-linear and ill-posed; the presence of the MIMO system partially overcomes limitations in recovery performanceswhile a further improvement might be achieved by considering multi-frequency sources.

The tests presented in this study considered air-matched propagation where the acoustic contrast between objects and background medium is very high. Future work will consider ultrasound tomographic experiments with water-matched cases, which is desirable for biomedical applications in which the information about the inner part of tissues is paramount.

## Figures and Tables

**Figure 1 sensors-22-07802-f001:**
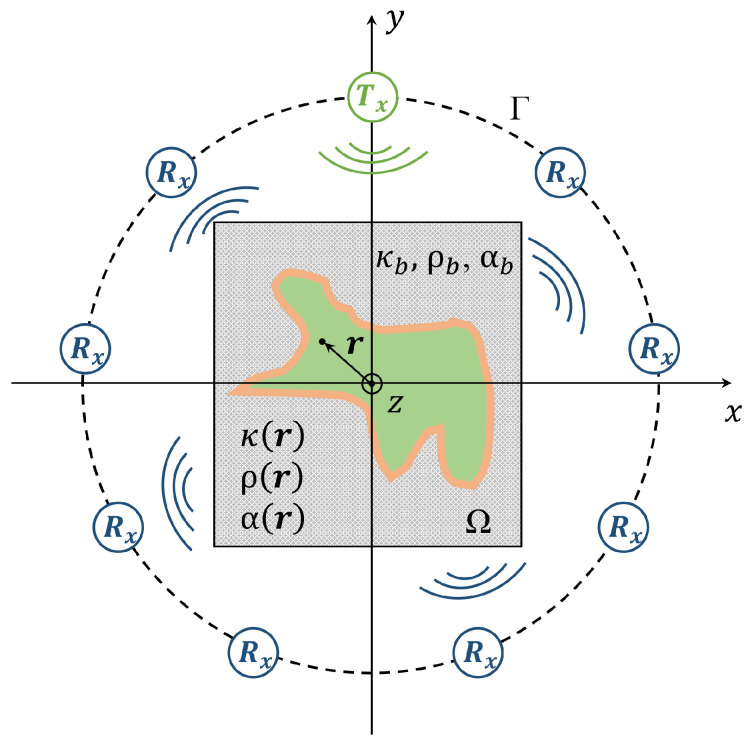
Sketch of the 2D ultrasound tomographic system. Γ is the curve containing ultrasound transducers. Ω is the region under investigation. The quantities κ,ρ and α are, respectively, the acoustic compressibility, density and attenuation of the objects located inside the background medium with parameters $\kappa_{b},\rho_{b}$ and $\alpha_{b}$.

**Figure 2 sensors-22-07802-f002:**
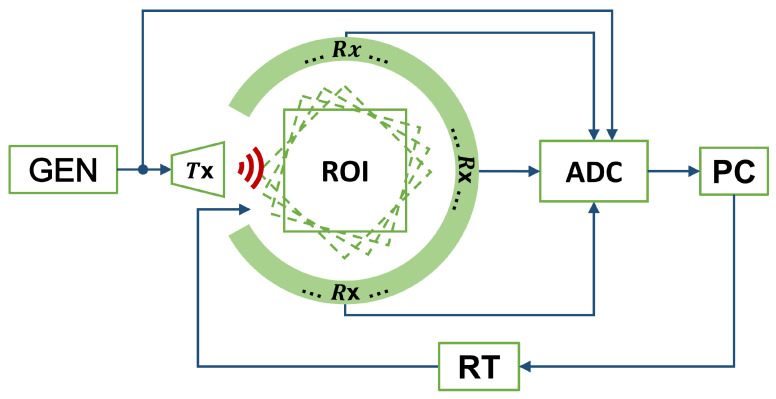
Block diagram of the ultrasound tomography system. The transmitter (Tx) is piloted by a waveform generator (GEN). The US waves impinge the region of interest (ROI) and the scattered waves are sensed by the receivers (Rx). Both transmitted and received waves are digitised by an analog-to-digital converter (ADC). This process is repeated for every angular position of the ROI, and the rotation is carried out via a rotating table (RT). The entire measurement process is managed by a personal computer (PC).

**Figure 3 sensors-22-07802-f003:**
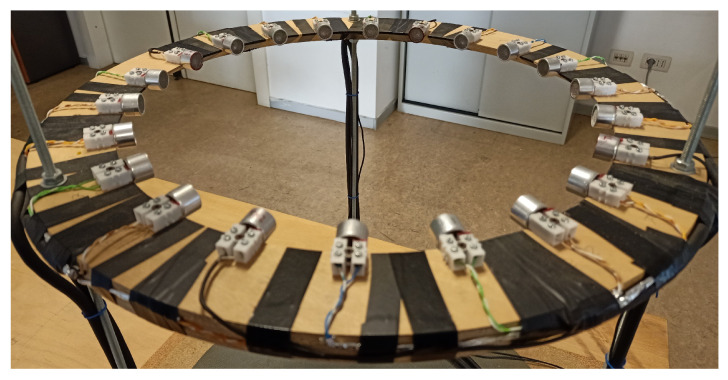
Picture of the US transducers circular array. The measurement curve is composed of one transmitter and 21 receivers.

**Figure 4 sensors-22-07802-f004:**
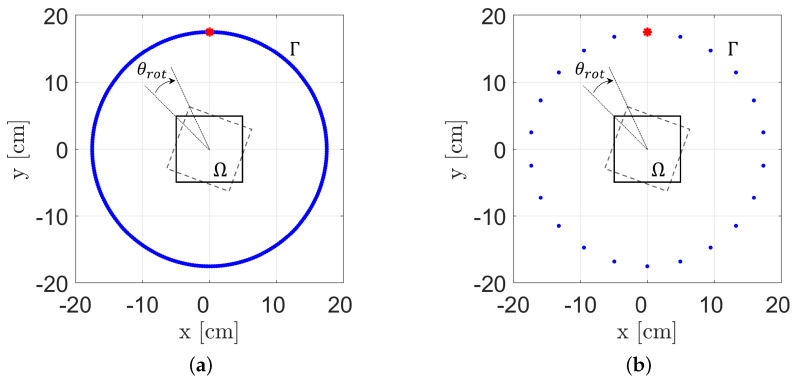
Measurement configurations adopted in the numerical and experimental analyses. In both cases, 308 view angles were considered by rotating the investigation domain, Ω, of equally spaced angular steps $\theta_{rot}=1.165$ by adopting 308 (**a**) and 21 (**b**) receivers.

**Figure 5 sensors-22-07802-f005:**
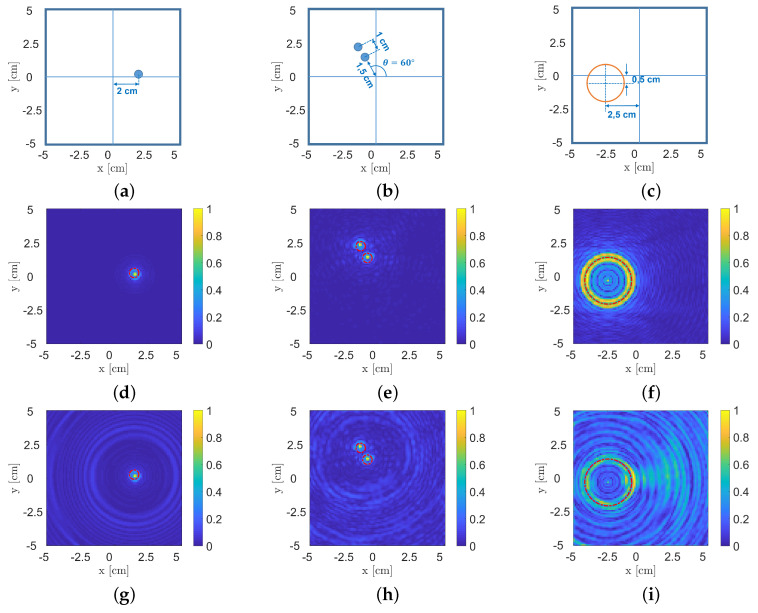
Test cases considered for the experimental campaign: (**a**) single 4 mm-diameter metallic cylinder, (**b**) a pair of 4 mm-diameter metallic cylinders and (**c**) a 3 cm-diameter hollow cardboard cylinder. (**d**–**f**) Support estimations obtained by processing corresponding numerical examples via the indicator reported in Equation (Equation 8) in the case of the measurement configuration in Figure 4a and (**g**–**i**) in case of the measurement configuration in Figure 4b. (**j**–**l**) Support estimations in the case of the experimental data collected by the in-house US system.

**Figure 6 sensors-22-07802-f006:**
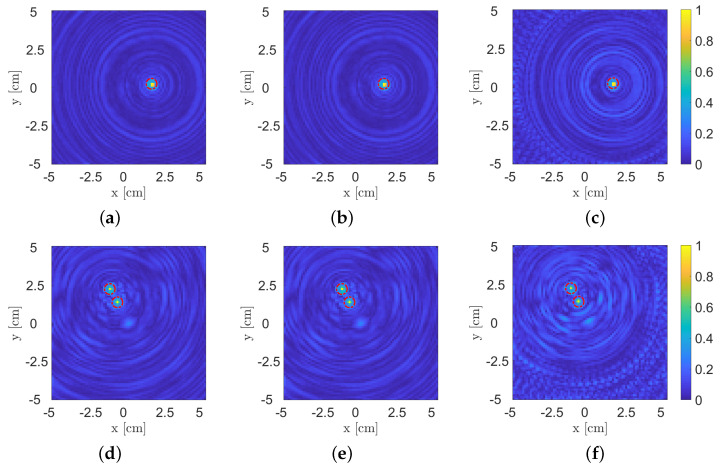
Experimental performance assessment in case of data reduction for the three scenarios acquired during the measurement campaign. (**a**,**d**,**g**) Full data processing (308 view angles), (**b**,**e**,**h**) 50% data reduction (154 view angles) and (**c**,**f**,**i**) 75% data reduction (77 view angles).

**Table 1 sensors-22-07802-t001:** Parameters of k-Wave simulations.

**Acquisition Time (single view)**	15 [ms]
**Sample Time**	1 [ns]
**Grid Resolution**	0.35 × 0.35 [mm^2^]
**Number of Pixels**	984 × 984
**PML Size**	7 [mm]

## Data Availability

All the data are available with prior request to the authors.

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
