# Peer review of "An Experimental Ultrasound System for Qualitative Tomographic Imaging"

_sensors, 2022, doi:10.3390/s22207802_

Round 1

Reviewer 1 Report

1. The focus of this article is an ultrasound system, as depicted in the title. However, the introduction presents many imaging algorithms but rare works on system design. It also lacks the necessary statement about the advances of the designed system, e.g., why is it developed, what's the superiority...Therefore, it still needs extensive work on reviewing the progress of ultrasound system design in the introduction. 

2. Details of the experiments are needed. For example, a 40kHz single-frequency signal is used for excitation. What's the duration time? In the k-wave simulation, how do you set the grid size, time step, boundary conditions, and so on?

3. Is there any data calibration for experimental imaging? How is it performed? How do you perform raw data to p_scat? The article mentioned a "beating process", what is it referred to? I hope the readers can learn how to design a system, record and process raw data step by step after reading this article, while this paper cannot provide sufficient information at this stage. 

Reviewer 2 Report

Suggestion: For future research, you could improve the materials for the experimental stage.

Reviewer 3 Report

Dear Authors,

Thank you for the possibility to get acquainte with Your paper. I found Your paper interesting. Here are my suggestions/questions:

- please consider improving some sentences with native speaker English

- I would shorten some parts of introduction and instead improve paragraphs on more sophisticated aspects

-  I believe in Introduction and Mathematical Formulation the complex terms could be more explained for reader

- I would suggest to relate the obtain results to other researchers 

Do You think of applying the device in medical studies as well ?

What was the background medium in Your experiment?

Round 2

Reviewer 1 Report

Thanks.